# Molecular Surveillance of Artemisinin-Resistant *Plasmodium falciparum* Parasites in Mining Areas of the Roraima Indigenous Territory in Brazil

**DOI:** 10.3390/ijerph21060679

**Published:** 2024-05-25

**Authors:** Jacqueline de Aguiar-Barros, Fabiana Granja, Rebecca de Abreu-Fernandes, Lucas Tavares de Queiroz, Daniel da Silva e Silva, Arthur Camurça Citó, Natália Ketrin Almeida-de-Oliveira Mocelin, Cláudio Tadeu Daniel-Ribeiro, Maria de Fátima Ferreira-da-Cruz

**Affiliations:** 1Malaria Control Center, Epidemiological Surveillance Department, General Health Surveillance Coordination, SESAU-RR, Roraima 69305-080, Brazil; barros.jacqueline@gmail.com; 2Postgraduate Program in Biodiversity and Biotechnology–BIONORTE Network/Roraima Federal University (UFRR), Roraima 69310-000, Brazil; fabiana.granja@ufrr.br; 3Biodiversity Research Centre, Roraima Federal University (UFRR), Roraima 69304-000, Brazil; danieldas.silva@yahoo.com; 4Graduate Program in Natural Resources, Federal University of Roraima (UFRR), Roraima 69304-000, Brazil; 5Laboratório de Pesquisa em Malária, Instituto Oswaldo Cruz, Fundação Oswaldo Cruz (Fiocruz), Rio de Janeiro 21040-360, Brazil; rebeccasantos@aluno.fiocruz.br (R.d.A.-F.); lucasqueiroz@aluno.fiocruz.br (L.T.d.Q.); nataliaketrin@gmail.com (N.K.A.-d.-O.M.); malaria@fiocruz.br (C.T.D.-R.); 6Centro de Pesquisa, Diagnóstico e Treinamento em Malária (CPD-Mal)/Reference Center for Malaria in the Extra-Amazon Region of the Brazilian Ministry of Health, Fiocruz, Rio de Janeiro 21040-900, Brazil; 7Research Support Center in Roraima (NAPRR) of the National Institute for Amazonian Research (INPA), Roraima 69301-150, Brazil; arthur.cito@inpa.gov.br

**Keywords:** Amazon, malaria, chemoresistance, *pfk13*, Guiana Shield

## Abstract

Multidrug- and artemisinin-resistant (ART-R) *Plasmodium falciparum (Pf)* parasites represent a challenge for malaria elimination worldwide. Molecular monitoring in the Kelch domain region *(pfk13)* gene allows tracking mutations in parasite resistance to artemisinin. The increase in illegal miners in the Roraima Yanomami indigenous land (YIL) could favor ART-R parasites. Thus, this study aimed to investigate ART-R in patients from illegal gold mining areas in the YIL of Roraima, Brazil. A questionnaire was conducted, and blood was collected from 48 patients diagnosed with *P. falciparum* or mixed malaria (*Pf + P. vivax*). The DNA was extracted and the *pfk13* gene was amplified by PCR. The amplicons were subjected to DNA-Sanger-sequencing and the entire amplified fragment was analyzed. Among the patients, 96% (46) were from illegal mining areas of the YIL. All parasite samples carried the wild-type genotypes/ART-sensitive phenotypes. These data reinforce the continued use of artemisinin-based combination therapies (ACTs) in Roraima, as well as the maintenance of systematic monitoring for early detection of parasite populations resistant to ART, mainly in regions with an intense flow of individuals from mining areas, such as the YIL. This is especially true when the achievement of falciparum malaria elimination in Brazil is planned and expected by 2030.

## 1. Introduction

*Plasmodium falciparum* is the world’s leading and most lethal cause of malaria, making it one of the main challenges to global public health. In 2022, it was responsible for 97% (249 million) of cases and 95% (608,000) of deaths from malaria worldwide [1].

In the Americas region, *P. falciparum* was responsible for 28% of the total 552,000 cases in 2022. Venezuela, Brazil, and Colombia accounted for around 73% of malaria cases in this region [1]. In Brazil, this parasite was responsible for 15.92% of the 142,522 cases reported in the country in 2023, according to data from the Malaria Epidemiological Surveillance Information System (Sivep-Malaria) [2].

Roraima is one of the states that make up the Brazilian Amazon, where 99% of malaria cases occur [3]. According to Sivep-Malaria, the proportion of *P. falciparum* malaria in the state was 29% of the total of 34.555 malaria cases reported in 2023 [2].

In the global strategy to combat malaria, the World Health Organization (WHO) has presented targets for 2016 to 2030, including reducing malaria cases by at least 90% by 2030 and eliminating malaria in at least 35 countries [4]. In this context, Brazil launched the *P. falciparum* malaria elimination plan in 2015 and proposed eliminating malaria by 2035 in 2022. One of the intermediate targets of the Brazilian National Plan is the elimination of *P. falciparum* malaria by 2030 [5].

A major obstacle to eliminating malaria is the emergence of *P. falciparum* multidrug resistance [6]. In fact, in 1960, only a decade after starting large-scale use of chloroquine, *P. falciparum* resistance to this drug was detected in Colombia, Cambodia, and Thailand, and it spread rapidly to other endemic countries, including Brazil [7]. In the 1970s, the combination of sulphadoxine and pyrimethamine (SP) was introduced to treat *P. falciparum* infections in South America, but chemoresistance was also soon reported in Colombia, Brazil, Peru, Venezuela, and Bolivia [8,9,10,11]. In the 1980s, mefloquine (MQ) was proposed in Brazil as a therapeutic alternative for non-severe malaria, but soon MQ-resistant parasites were reported [12].

Following the reports of treatment failure and the spread of multidrug-resistant *P. falciparum* parasites, the WHO recommended ART-based combination therapies (ACTs) to treat uncomplicated *P. falciparum* malaria [13]. In addition, the use of ART as monotherapy for malaria treatment was suspended to prevent the emergence of drug resistance [14]. ACTs combine two active substances with different mechanisms of action: an ART derivative and another antimalarial drug. The first has a short plasma half-life (1 to 2 h) and aims to rapidly reduce the parasite biomass. The second antimalarial drug has a longer plasma half-life (days to weeks) and is designed to eliminate the remaining parasites [6,15].

The Brazilian National Malaria Control Program (PNCM) introduced treatment with ACTs for uncomplicated *P. falciparum* malaria in 2005, which led to an increasing decline in malaria in the Brazilian Amazon region [16]. The same epidemiological dynamic can be observed in other endemic countries worldwide [17].

The detection of ART resistance in Southeast Asia, first in Cambodia in 2008 and then in China, Vietnam, Thailand, and Myanmar, has highlighted an obstacle in the global malaria elimination effort [18]. The emergence of this resistance can be attributed to monotherapy with unregulated ART or artesunate (AS), which had been available in this region since the mid-1970s, as well as the availability of these drugs in the private health sector [19].

In 2010, a mutation associated with ART resistance was identified in Guyana, a South American country. Genomic analyses indicate that the mutation in Guyana did not spread from Southeast Asia but occurred independently [20].

The state of Roraima, together with the western part of Amapá, the northern Amazon, and Pará in Brazil, and the territories of Guyana, Suriname, French Guiana, Venezuela, and Colombia, is part of the Guiana Shield. This region is considered a potential source for the emergence of malaria resistance in South America, as the subsoil rich in gold and other minerals attracts prospectors to the indigenous forest areas and, therefore, is increasing the size of the human population [21,22,23].

In the last decade, the proportion of *P. falciparum* infections in Roraima has increased significantly, probably due to delayed diagnosis and treatment of Venezuelan migrants and gold miners living in illegal mining areas in the YIL [24]. In addition, Venezuela’s economic collapse led to a shortage of antimalarial drugs and the search for treatment in neighboring countries, particularly Brazil, increasing imported malaria cases, including *P. falciparum* infections [25].

In Roraima, there was a 44% rise in malaria cases in 2020 [24]. In the same year, illegal mining in the YIL increased by 30%. More than half (52%) of the mines in this region are located on the Uraricoera River, but mines have also been found on the banks of the Mucajaí, Couto de Magalhães, Parima, and Catrimani rivers [26].

The mines are located in isolated forest areas without access to diagnosis and treatment by the Brazilian Unified Health System (SUS). For this reason, miners with a fever tend to take antimalarials of dubious or even illegal origin to avoid having to stop mining. This scenario may favor the selection of ART-resistant parasites and underlines the need for molecular surveillance of antimalarial resistance in this region [24,27,28].

Surveillance of molecular markers of antimalarial resistance is an important strategy for detecting treatment failure and should be implemented to detect resistant parasites early and prevent their spread [29,30,31]. In 2014, mutations in the propeller domain of the Kelch 13 gene on chromosome 13 (*pfk13*) were associated with delayed elimination of parasites in vitro and in vivo, and this molecular marker has been used for global surveillance of artemisinin resistance (ART-R) [6,32]. The following mutations have been validated by in vitro and in vivo studies: C580**Y**, R561**H**, R539**T**, I543**T**, P553**L**, M476**I**, N458**Y**, Y493**H**, F446**I**, and P574**L**. The mutations P441**L**, T449**A**, C469**F/Y**, A481**V**, R515**K**, P527**H**, N537**I/D**, G538**V**, V568**G**, R622**I**, and A678**V** are considered ART-R molecular markers. In addition, less common *pfk13* variants such as K479**I**, G533**A**, R575**K**, M579**I**, D584**V**, P667**T**, F673**I**, and H719**N** have also been associated with delayed parasite elimination [6].

Considering that the future efficacy of ACTs is endangered by the emergence of resistance artemisinin, this study aims to investigate single nucleotide polymorphisms (SNPs) associated with *P. falciparum* ART-R in the *pfk13* gene in the state of Roraima.

## 2. Materials and Methods

The study site was the municipality of Boa Vista (2°49′10″ N and 60°40′23″ W, the capital of Roraima, which is located in the far north of Brazil and is the only capital above the equator [33]. Almost the entire Roraima state is part of the Guiana Shield, along with the western part of Amapá and the northern parts of the Amazonas and Pará in Brazil. In addition to Brazil, Guyana, Suriname, French Guiana, southern Venezuela, and eastern Colombia comprise the Guiana Shield. There is an intense migratory flow of miners in this Shield and C580**Y** and R539**T** mutations in the *pfk13* gene have already been identified in Guyana [20] (Figure 1).

According to estimates by the Brazilian Institute of Geography and Statistics (IBGE), the city currently has 436,591 inhabitants and is home to 67% of the state’s population. It borders the municipalities of Normandia, Pacaraima, and Amajari to the north; Mucajaí and Alto Alegre to the south; Bonfim, Cantá, and Normandia to the east; and the municipality of Alto Alegre to the west.

Boa Vista is home to the state’s most sought-after public health institutions. The two health facilities with the highest number of malaria reports, according to Sivep-Malaria, were selected as sample collection sites: Emergency Service Cosme e Silva and Basic Health Unit Sayonara Maria Dantas, located in the West Zone of the city (Figure 2). This urban area has the highest employment density, 70% of the city’s 56 neighborhoods, and most of the population has a low monthly income. The West Zone emerged in the 1990s due to interregional migration flows, rural exodus and gold prospectors who moved to Boa Vista after mining was banned in the early 1990s [34]. The samples were collected from December 2021 to June 2022, during the transition period from the rainy to the dry season, when the mosquito population increases seasonally [34].

This study was approved by the Research Ethics Committee of the Federal University of Roraima (CEP/UFRR): CAAE 24122619.6.0000.5302, on 17 March 2020. Participants signed an informed consent form (ICF) to participate in this study. Individuals over the age of 18 who had been diagnosed with *P. falciparum* or mixed malaria (*P. vivax* + *P. falciparum*) by a thick blood smear were included. Those excluded from this study were children under 18, indigenous villagers, individuals who could not read, and those who refused to sign the informed consent form. Participants in this study were asked questions related to the individual and the disease. Blood was collected by venipuncture of 5 mL of peripheral blood. Part of the blood (around 50 microliters) was transferred directly from the syringe to filter paper (Whatman 903 Protein Saver Cards, Merck (Sigma), Darmstadt, Germany) and the rest was placed in a vacutainer tube containing EDTA (Becton, Dickinson & Company, Franklin Lakes, NJ, USA).

The samples were transported to the Molecular Biology Laboratory of the Biodiversity Research Center of the Roraima Federal University. The tubes containing blood were centrifuged at 3000× *g* for 10 min to remove the plasma. The “red blood cell concentrate” (containing leukocytes and platelets) was added to the cryopreservation solution glycerolyte 57 (Baxter Inc., Illinois, USA) volume by volume (*v*/*v*), followed by the aliquoting of each sample. The aliquots with the cryopreservation solution were stored at −20 °C until DNA extraction.

Deoxyribonucleic acid (DNA) was extracted using the column technique (centrifugation method), using the QIAamp DNA blood mini kit (Qiagen, Hilden, Germany), according to the manufacturer’s instructions, from 500 µL of the “red blood cell concentrate”.

The *pfk13* gene fragment was amplified by nested polymerase chain reaction (PCR) [35]. The following primers were used in the first reaction: K13 F: 5′CGGAGTGACCAAATCTGGGA3′ and K13 R: 5′GGGAATCTGGTGGTAACAGC3′.

A mixture of PCR reagents was prepared with 13.75 µL of ultrapure water, 5 µL Taq 5× hotfirepol^®^, (Solis, Tartu, Estonia), 0.625 µL of each of the primers (10 pmol), and 5 µL of the *P. falciparum* DNA sample was added to each mixture. The PCR conditions in the thermal cycler were as follows: initialization at 95 °C for 15 min; 30 cycles of denaturation at 95 °C for 30 s, annealing of the primers at 58 °C for 2 min, and extension at 72 °C for 2 min; and final elongation at 72 °C for 10 min.

The following primers were used for the second PCR: K13 N_F: 5′GCCTTGTTGAAAGAAGCAGA3′ and K13 N_R: 5′GCCAAGCTGCCATTCATTTG3′. A mixture was prepared with 37.5 µL ultrapure water, 10 µL Taq 5× hotfirepol^®^, 1.25 µL of each of the primers (10 pmol), and 5 µL of the first PCR amplified product was added to each mixture. The PCR conditions were as follows: initialization at 95 °C for 15 min; 40 cycles of denaturation at 95 °C for 30 s; annealing at 60 °C for 1 min and extension at 72 °C for 1 min; and final elongation at 72 °C for 10 min.

The final 849 base pairs (bp) amplified product (amplicon) was analyzed after electrophoresis in a 2% agarose gel, stained with Bluegreen (LGC), using a DNrBio-Imagining System/Model: MiniBIS Pro (UV) photo-documentation system (DNR, Jerusalem, Israel). A 100 bp molecular weight marker was used to check the size of the fragments (Figure 3).

The amplicons were purified using the Wizard^®^ Kit, according to the manufacturer’s instructions. The sequencing reaction was carried out using the Big Dye kit^®^ Terminator Cycle Sequencing Ready Reaction version 3.1 (Applied Biosystems, Carlsbad, CA, USA), according to the manufacturer’s instructions. The amplicons were subjected to Sanger-type sequencing using capillary electrophoresis on the ABI PRISM DNA Analyzer 3730 (Applied Biosystems, Carlsbad, CA, USA) of the PDTIS/Fiocruz genomic platform and ABI PRISM DNA Analyzer 3500 (Applied Biosystems, Carlsbad, CA, USA) of the LabMol/CBio/UFRR. 

The entire amplified fragment DNA sequence (codons 427 to 709) was analyzed using the ClustalW multiple sequence aligner in the BioEdit^®^ version 7.7.1 software (North Carolina State University, Raleigh, NC, USA), using the prototype 3D7 as the reference sequence (GenBank PF3D7_1343700: Reference 000002765 *Plasmodium falciparum* Genome Sequencing Consortium, release date: PlasmoDB # 29, 2016-october-12, available in https://www.ncbi.nlm.nih.gov/nuccore/).

The maps were created using the QGIS program version 3.28.10. Mining areas in Roraima were obtained from Mapbiomas [36]. The geopolitical limits of Brazil and the indigenous lands were accessed on the IBGE website [37].

## 3. Results

A total of 53 samples were collected, 42 from *P. falciparum* and 11 from mixed *P. falciparum* + *P. vivax* malaria infections. Of the total samples collected, 83% (44) were from men, ages 18 to 55, with a median of 36 years. Regarding the main activity performed 15 days before the symptoms, 96% (51) of participants reported mining and only 4% (02) agriculture. Detailed epidemiological information on these patients has recently been published [38]. 

Approximately 97% (48) of the samples were amplified by nested PCR for the *pfk13* gene, and all amplicons were purified and sequenced. Of the total sequenced samples, 46 (96%) were from patients reporting mining activity, and only 2 (4%) were from those who reported agricultural activities. The mines were mainly located in the YIL in the municipalities of Alto Alegre (40) and Mucajaí (5) (Figure 4). In addition, one participant came from gold mines in Itaituba/PA (1). The patients who reported working in agriculture were likely infected in the municipalities of Caracaraí (1) and Alto Alegre (1).

After aligning the nucleotide sample sequences with the 3D7 *P. falciparum* reference genome strain (PF3D7_1343700), no mutations were found in the *pfk13* gene, neither in the codons associated with ART-R nor in the other codons analyzed. Thus, in the amplified fragment, all *P. falciparum* sequence samples were identical to the wild-type 3D7 reference sequence (Figure 5).

## 4. Discussion

It is known that gold miners are considered at risk for malaria and that *P. falciparum* is widely distributed in several gold mining regions, especially in the Guiana Shield [23,39], where conditions are favorable for the selection of antimalarial-resistant parasites.

Slow elimination of the parasite after treatment with ACT characterizes partial resistance to ART. In the Brazilian endemic areas, parasites in the blood on day 3 (D3) after the start of therapy (D0) are rare in patients treated with ACT. Therefore, assessing *P. falciparum* clearance on D3 after initiation of treatment is an important parameter for monitoring ART-R [6]. However, in the present study, participants’ parasitemia was not assessed on day D3 after starting treatment with ACT because it was difficult to follow the vast majority of participants, who were miners that traveled to Boa Vista only for diagnosis and returned to the mines in YIL just after receiving the medication.

In recent years, illegal mining has increased in Roraima in the YIL, mainly in Mucajaí, Uraricoera, Catrimani, and Parima river areas. More than half (52%) of the total area damaged by mining is concentrated along the Uraricoera River [26]. Access routes to the mining areas are mainly via the rivers and forest areas of the municipalities of Alto Alegre, Amajari, Mucajaí, Caracaraí, and Iracema, or by plane via clandestine airstrips hidden in rural areas. The gold miners in this region do not have access to malaria diagnosis and treatment through the SUS, which is exclusively for the indigenous peoples living there [24]. Therefore, to avoid losing working days through travel to distant cities, gold miners buy the drug Artecom^®^ (dihydroartemisinin-piperaquine; Chongqing Tonghe Pharmaceutical Co. Ltd., Chongqing, China), which is not registered in Brazil and enters illegally through the Suriname, Guyana and French Guiana borders. They claimed that a single dose of this drug relieves the symptoms for a few days even though it cannot clear all the parasites. Then, this indiscriminate drug use makes miners a risk group for selecting *P. falciparum* parasites resistant to ART [39].

Currently, the only treatment for *P. falciparum* malaria is the ACTs. Consequently, the emergence of resistance to ART would have a devastating impact on the global goal of malaria elimination. It could also lead to the risk of selecting resistant parasites against partner drugs [6,40]. 

Molecular surveillance of *pfk13* helix domain polymorphism in endemic countries can play an important role in early warning of ART-R parasites [6,23,41,42]. The C580**Y** mutation in the *pfk13* gene is relevant for molecular surveillance of ART resistance in Southeast Asia [18,43]. This mutation is present in the vast majority of resistant parasites in Cambodia and reaches a prevalence of up to 70% at the border between Thailand and Myanmar [6]. In 2010, the C580**Y** mutation was also identified in 5% (5/98) of parasite samples from Guyana [20]. The appearance of this mutation in South America raised the question of whether there would be a risk of mutant alleles spreading to different hotspots in the Amazon basin through intensive migration across the Venezuela, Guyana, and Brazil borders as a result of illegal gold mining. However, between 2016 and 2017, the prevalence of this allele decreased to 1.6% (14/864) [42]. The in vitro competitive co-cultivation of *pfk13* mutant (C580**Y** and R539**T**) and non-mutant parasites from Guyana showed that the mutants have a growth deficit compared to the non-mutant wild parasites [42]. This disadvantage in the fitness of the ART-R mutant parasite could be one of the reasons why these *pfk13* gene mutants have not been fixed and, thus, have not spread from Guyana to neighboring countries in recent years [42]. However, we must bear in mind that the risk of the spread of such mutants in the Amazon basin, especially those from the Guiana Shield, is proportional to the intensity of mining activities whose main migration patterns include Venezuela, Guyana, and the state of Roraima in Brazil [23].

The World Health Organization drew attention to the presence of *P. falciparum* parasites carrying the C580**Y** mutation in infected Chinese travelers in Equatorial Guinea and Ghana upon their return to their country and hypothesized that these mutations were likely to have originated in Africa rather than Southwest Asia, but there was no evidence that this mutation had spread to parasite populations in African areas [6,44]. In fact, non-synonymous *pfk13* mutations are present at low frequencies in Africa [40,45]. Of the 35 non-synonymous mutations detected on that continent, the R561**H** was the most frequent, followed by A578S, which appears to be the most prevalent worldwide [40,45,46,47]. The R539**T** and P553**L** variants were identified in a sample from Angola, and the M476**I** in a sample from Equatorial Guinea [46]. R561**H**, M579**I**, and C580**Y** mutations can confer in vitro artemisinin resistance in African parasites [45].

Although 97% of the samples in our study came from individuals infected in mining areas, all sequences analyzed showed non-mutated/wild-type genotypes in the helix–loop domain of *pfk13*. This finding corroborates a study carried out in Roraima, which also described the absence of polymorphisms in the *pfk13* gene in parasite samples collected between 2016 and 2017 in the municipalities of Boa Vista, Pacaraima, and Rorainópolis [48].

Samples of *P. falciparum* from the Brazilian Amazon basin collected between 1984 and 2011, i.e., in the period before and after the introduction of ART treatment in Brazil, also showed no mutations in the *pfk13* gene [41], as reported in more recent studies in this region [49,50].

In 2013, the mutant *pfk13* A481**V** parasite, which is associated with delayed elimination of *P. falciparum* in patients in Southeast Asia [51], was detected in one of the 575 samples of *P. falciparum* malaria from Manaus/Brazil, but without clinical signs of ART-R [23]. The A504**D** mutation observed in 2018 in Colombia also did not lead to ART-R phenotypes [51]. Thus, further work is needed before these mutations can be considered markers for chemoresistance to ART in samples from South America.

As the P413**A** mutation in the BTB/POZ domain of the *pfk13* gene was recently identified in parasites from an African isolate of *P. falciparum* subjected to ART in vitro pressure [52], we intend to investigate mutations in the BTB/POZ domain, in addition to those here investigated in the *pfk13* gene helix domain [53].

Finally, considering that the *pfCoronin* gene is the main driver of reduced susceptibility to ART in Senegalese parasites developed in vitro [54], we intend to extend our investigations to this gene, due to the possibility of synergism with *pfk13* in cases of partial resistance to ART [54,55].

## 5. Conclusions

Given the results of this study, treatment with ART derivatives can continue to be used as the first-line treatment for *P. falciparum* malaria in Brazil. We recommend continuing molecular surveillance to track ART resistance in Roraima, as mutant parasites could be introduced and/or selected due to the influx of miners into the Guiana Shield, which consists of Brazil, French Guiana, Suriname, Guyana, Venezuela, and Colombia.

## Figures and Tables

**Figure 1 ijerph-21-00679-f001:**
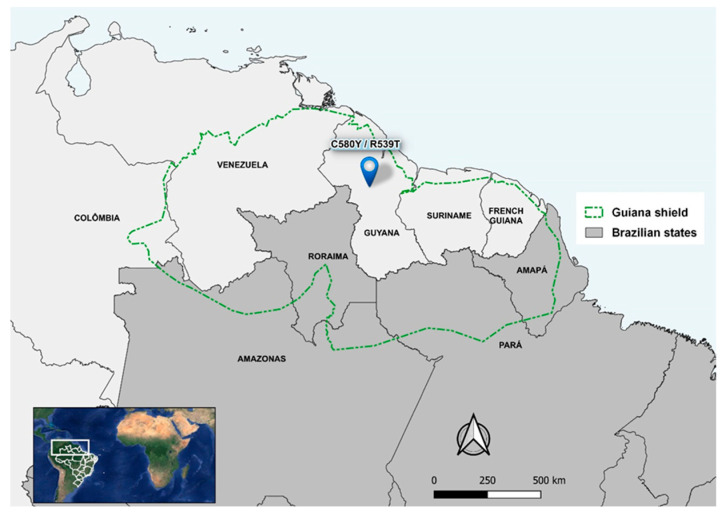
Location of Roraima in the Guiana Shield and mutations in the *pfk13* gene (C580**Y** and R539**T**) in Guyana. Maps were produced by the authors.

**Figure 2 ijerph-21-00679-f002:**
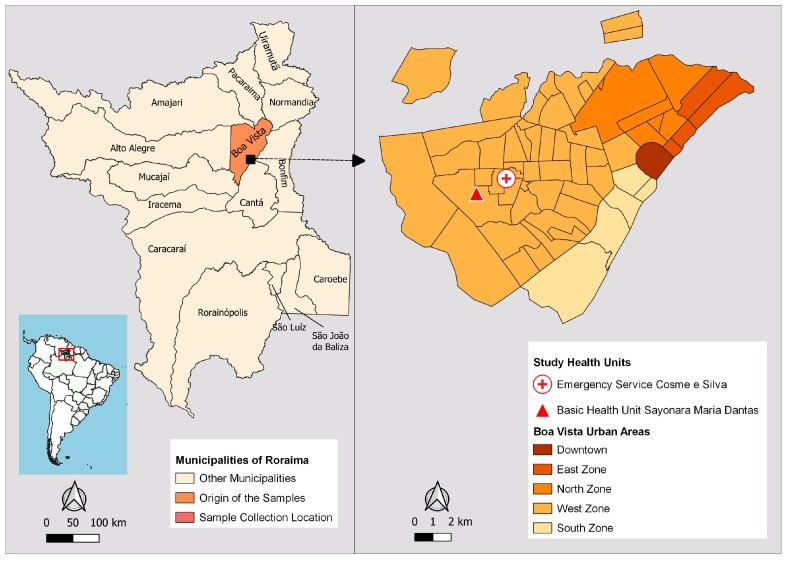
Maps showing the location of Boa Vista, the urban areas, and the study areas. The maps were created by the authors.

**Figure 3 ijerph-21-00679-f003:**
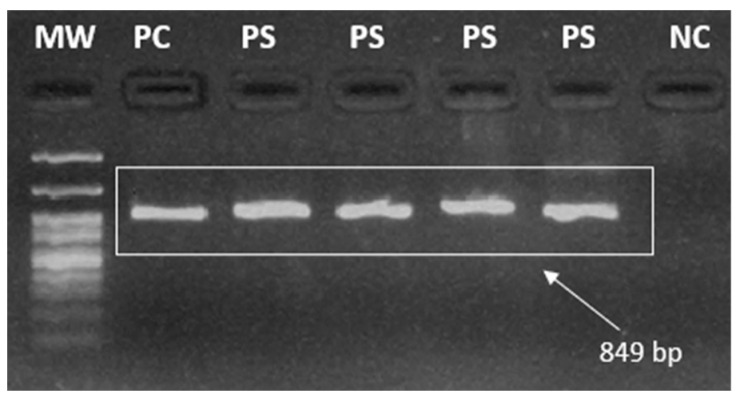
Representative illustration of a 2% agarose gel highlighting the amplicons from *pfk13* nested PCR. Legend: MW = Molecular Weight (100 bp), PC = Positive Control, PS = Positive Sample, NC = Negative Control. Image taken by the authors.

**Figure 4 ijerph-21-00679-f004:**
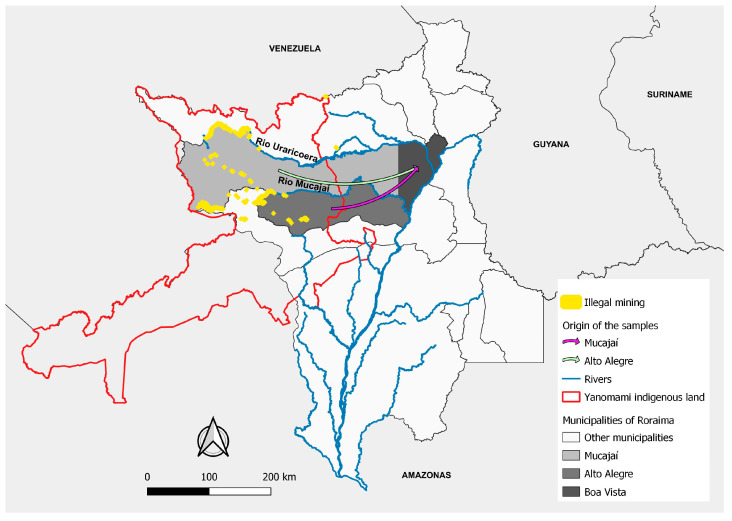
Origins of the samples, locations of the mines, rivers, Yanomami indigenous land, municipalities of infection, the Guiana Shield, and the study site. Map made by the authors.

**Figure 5 ijerph-21-00679-f005:**
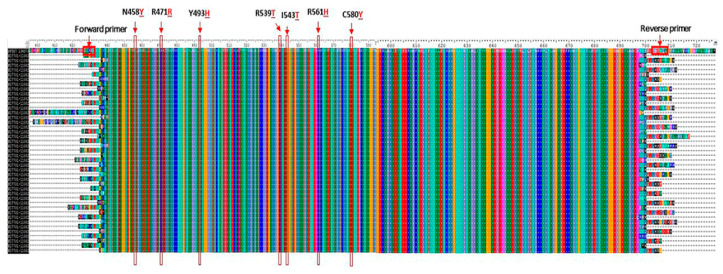
Alignment of *pfk13* gene sample sequences. SNPs associated with ART resistance are highlighted in red. The red rectangle at the beginning of the sequence corresponds to the primer sequences. Each color represents one amino acid.

## Data Availability

The original data (sequences) presented in the study are openly available (deposited) in Genbank™ with accession numbers PP584057-PP584104.

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
