# Peer review of "Molecular Surveillance of Artemisinin-Resistant Plasmodium falciparum Parasites in Mining Areas of the Roraima Indigenous Territory in Brazil"

_ijerph, 2024, doi:10.3390/ijerph21060679_

Round 1

Reviewer 1 Report

Comments and Suggestions for Authors

This article addresses an important topic regarding surveillance of malaria drug resistance in endemic area. I appreciated the description of the socio-economic parameters that guided this study. It is very interesting to put the risks of emergence of drug resistance into perspective with human behavior (taking of medicines, population migrations, etc). The study and article are well constructed, the strategy clearly explained and I recommend the publication of the article after minor revisions.

-Line 41: the reference “2” is in superscript but in square bracket in the rest of the document.

-Line 60: the reference “12” is only available in Portuguese, please add an equivalent reference in English.

Please clarified the date for the first use of artemisinin (in monotherapy and/or in combination) in Brazil (in line 58 the authors mentioned 1980’s but 2005 in line 70).

The reference 18 and 43 are the same.

The figure 5 need to be modified because unreadable even at maximum zoom.

In line 273, the authors referred to development of complete resistance to ART. To date ART-resistance is always demonstrated to be partial (artemisinin still kill the main part of a resistant parasite population even clonal) both in vivo and in vitro. The major risk is the selection of parasites resistant to both artemisinin and partner drugs.

The authors should update the paragraph line 300- 306 regarding the current situation in Africa. Pfk13 mutations are no more “rare” in Africa and many publications in 2023-2024 explained it. The R561H mutations should be also mentioned.

-Line 325: the reference “51” is not suitable for the P413A mutation (Paloque L, Coppée R, Stokes BH, Gnädig NF, Niaré K, Augereau JM, Fidock DA, Clain J, Benoit-Vical F. Mutation in the Plasmodium falciparum BTB/POZ Domain of K13 Protein Confers Artemisinin Resistance. Antimicrob Agents Chemother. 2022 Jan 18;66(1):e0132021. doi: 10.1128/AAC.01320-21. Epub 2021 Oct 4. PMID: 34606334; PMCID: PMC8765297.)

-Line 328: the following reference is missing regarding the link between Pfcoronin mutations and artemisinin resistance (Demas AR, Sharma AI, Wong W, Early AM, Redmond S, Bopp S, Neafsey DE, Volkman SK, Hartl DL, Wirth DF. Mutations in Plasmodium falciparum actin-binding protein coronin confer reduced artemisinin susceptibility. Proc Natl Acad Sci U S A. 2018 Dec 11;115(50):12799-12804. doi: 10.1073/pnas.1812317115. Epub 2018 Nov 12. PMID: 30420498; PMCID: PMC6294886.)

Comments on the Quality of English Language

The language may be improved, for example in line 322 “quimioresistance” must be replaced by “chemoresistance”.

Author Response

We thank the reviewer for the suggestions and comments that improve the MS.

  • Line 41: the reference “2” is in superscript but in the square bracket in the rest of the document.

The Superscript was removed at reference 2.

  • Line 60: the reference “12” is only available in Portuguese, please add an equivalent reference in English.

The reference “12” was replaced with a more adequate one.

  • Please clarify the date for the first use of artemisinin (in monotherapy and/or in combination) in Brazil (in line 58, the authors mentioned the 1980’s, but 2005 in line 70).

The Brazilian Malaria Program officially recommended using ACTs in 2005; some places had locally used ART mono or combined before the official recommendation. Thus, the sentences on lines 58, 59, and 60 were modified for better understanding.

  • The references 18 and 43 are the same.

The numbers were now corrected and the repeated reference was deleted.

  • Figure 5 needs to be modified because it is unreadable even at maximum zoom.

Figure 5 was modified for a better 300dpis resolution.

  • In line 273, the authors referred to development of complete resistance to ART. To date ART-resistance is always demonstrated to be partial (artemisinin still kill the main part of a resistant parasite population even clonal) both in vivo and in vitro. The major risk is the selection of parasites resistant to both artemisinin and partner drugs.

This sentence was modified as requested; please see lines 270 and 271.

  • The authors should update the paragraph line 300- 306 regarding the current situation in Africa. Pfk13 mutations are no more “rare” in Africa and many publications in 2023-2024 explained it. The R561H mutations should be also mentioned.T

The paragraph was updated, including the R561H mutation, as suggested. Please see lines 296 - 302

  • Line 325: the reference “51” is not suitable for the P413A mutation (Paloque L, Coppée R, Stokes BH, Gnädig NF, Niaré K, Augereau JM, Fidock DA, Clain J, Benoit-Vical F. Mutation in the Plasmodium falciparum BTB/POZ Domain of K13 Protein Confers Artemisinin Resistance. Antimicrob Agents Chemother. 2022 Jan 18;66(1):e0132021. doi: 10.1128/AAC.01320-21. Epub 2021 Oct 4. PMID: 34606334; PMCID: PMC8765297.)

Reference 51 was replaced by a more adequate one and numbered 53.

Line 328: the following reference is missing regarding the link between Pfcoronin mutations and artemisinin resistance (Demas AR, Sharma AI, Wong W, Early AM, Redmond S, Bopp S, Neafsey DE, Volkman SK, Hartl DL, Wirth DF. Mutations in Plasmodium falciparum actin-binding protein coronin confer reduced artemisinin susceptibility. Proc Natl Acad Sci U S A. 2018 Dec 11;115(50):12799-12804. doi: 10.1073/pnas.1812317115. Epub 2018 Nov 12. PMID: 30420498; PMCID: PMC6294886.)

This reference was added, as suggested.

Comments on the Quality of English Language

The language may be improved, for example in line 322 “quimioresistance” must be replaced by “chemoresistance”.

The language was improved throughout the MS.

Reviewer 2 Report

Comments and Suggestions for Authors

De-Aguiar-Barros et al present a very important piece of work very relevant to the elimination of malaria. Molecular surveillance in order to identify SNPs associated with P. falciparum ART-R in the pfk13 gene is needed to ensure global elimination of malaria.

Some minor comments/observations;

1. What does ACT in the abstract stand for?

2. what is SP? line 56

3. I do not understand the use of the phrase, "probably for delay" as used in line 90

Author Response

We thank the reviewer for the suggestions and comments that certainly improve the MS.

  1. What does ACT in the abstract stand for?

A sentence on this sense was inserted in lines 28-29 of the abstract session.

  1. what is SP? line 56

SP is the combination of sulfadoxine (SP) and pyrimethamine (P). This statement was introduced in line 56

  1. I do not understand the use of the phrase, "probably for delay" as used in line 90

The sentence in line 90 was replaced by “…probably due to delayed diagnosis and treatment….”